# Ivermectin for Prophylaxis and Treatment of COVID-19: A Systematic Review and Meta-Analysis

**DOI:** 10.3390/diagnostics11091645

**Published:** 2021-09-08

**Authors:** Mario Cruciani, Ilaria Pati, Francesca Masiello, Marina Malena, Simonetta Pupella, Vincenzo De Angelis

**Affiliations:** 1Italian National Blood Centre, National Institute of Health, 00162 Rome, Italy; ilaria.pati@iss.it (I.P.); francesca.masiello@iss.it (F.M.); simonetta.pupella@iss.it (S.P.); vincenzo.deangelis@iss.it (V.D.A.); 2Infectious Diseases Unit, AULSS9 Scaligera, 37100 Verona, Italy; marina.malena@aulss9.veneto.it

**Keywords:** ivermectin, SARS-CoV-2, COVID-19, systematic review, meta-analysis

## Abstract

Background. Ivermectin has received particular attention as a potential treatment for COVID-19. However, the evidence to support its clinical efficacy is controversial. Objectives. We undertook a new systematic review of ivermectin for the treatment and prophylaxis of COVID-19, including new primary studies, outcomes other than mortality, and grading the quality of the available evidence following the Cochrane guidance for methodology. Methods. We searched electronic databases, repository databases, and clinical trial registries (up to June 2021). The measure of treatment effect was risk difference (RD) with 95% confidence intervals (CIs). The GRADE system was used to assess the certainty of the evidence. Results. The review includes 11 RCTs (2436 participants). The certainty of the available evidence was quite low or very low due to risk of bias, inconsistency, and imprecision. When the analysis was limited to patients with baseline mild or moderate disease (8 reports, 1283 patients), there were no differences in mortality between ivermectin and control groups (low level of certainty); in patients with baseline severe diseases (3 reports, 304 patients), the use of ivermectin significantly decreased mortality compared to the controls (RD −0.17; 95% CIs, −0.24/−0.10; *p* = 0.00001; low level of certainty). In terms of disease progression (to severe pneumonia, admission to intensive care unit, and/or mechanical ventilation), the results were much the same. At day 14, the rate of patients with a negative RT-PCR test was 21% higher (from 5 to 36% higher) for ivermectin recipients than it was for the controls (low quality of evidence). Three studies (736 subjects) indicated that prophylaxis with ivermectin increased the likelihood of preventing COVID-19 compared to controls (low quality of evidence). Serious adverse events were rarely reported. Conclusions. There is limited evidence for the benefit of ivermectin for COVID-19 treatment and prophylaxis, and most of this evidence is of low quality. Further evidence is needed to fine-tune potential indications and optimal treatment protocols for ivermectin as a treatment for COVID-19.

## 1. Background

Several drugs have been considered for the treatment of SARS-CoV-2, and various unconventional treatments have been hailed as potential cures for COVID-19 [1,2,3,4]. One of the latest putative silver bullets against COVID-19 is ivermectin [5,6,7]. Apart from its invaluable therapeutic role in parasitic disease such as onchocerciasis and strongyloidiasis [8], there is also an increasing body of evidence showing the potential of ivermectin as an antiviral agent [9,10,11,12]. Recently, Caly et al. reported on the antiviral activity of ivermectin against SARS-CoV-2 [13]. In addition to its antiviral activity, ivermectin has proven to have anti-inflammatory effects [14,15]. Although the basis of its anti-inflammatory activity remains unclear, it has been suggested that this phenomenon is closely related to the clinical utility of ivermectin in the cytokine storm phase of COVID-19 [16].

In the past year, ivermectin has received special attention as a potential drug for the treatment and prophylaxis of COVID-19. Indeed, a number of clinical studies have been conducted in various countries, and there is a remarkable number of trials registered with ClinicalTrials.gov and with other clinical trials registries [6]. Recently, the number of primary studies and systematic reviews/meta-analysis focusing on ivermectin for the management of COVID-19 has increased substantially [6,8,17,18]. However, the evidence of the clinical efficacy of ivermectin is controversial [19]. Some of these reviews conclude that ivermectin reduced mortality compared to the standard treatment, but no other outcomes (clinical or virological) were analyzed [6,17,18]. In the systematic reviews available, we observed variability regarding the types and number of studies selected and included, as well as differences in the eligibility criteria and statistical methods used; even more importantly, the assessment of the methodological quality of the studies included and of the quality of evidence, key methodological procedures when conducting systematic reviews, was carried out infrequently with the potential for spurious or fallacious findings [20]. Therefore, we undertook a new systematic review of ivermectin for the treatment and prophylaxis of COVID-19, including new primary studies, outcomes other than mortality, and grading the quality of the available evidence following the Cochrane guidance for methodology.

## 2. Materials and Methods

This systematic review was conducted according to recommended PRISMA checklist guidelines (Appendix A) [21]. The protocol is registered on PROSPERO (registration number CRD42021256414).

### 2.1. Search Strategy

A computer-assisted literature search of the MEDLINE (through PUBMED), EMBASE, SCOPUS, OVID, and Cochrane Library electronic databases was carried out (latest search 1 June 2021) to identify clinical trials for the use of ivermectin for COVID-19. A combination of the following text words and MeSH terms was used: COVID-19/SARS- CoV-2 AND ivermectin. We also searched preprint repository databases (medRxiv, bioRxiv) and clinical trial registries (clinicaltrials.gov, who.int/clin.gov, accessed on 1 June 2021) for study details and study results. In addition, we checked the reference lists of the most relevant items (original studies and reviews) in order to identify potentially eligible studies not captured by the initial literature search.

### 2.2. Study Selection and Inclusion Criteria

Studies were selected independently by two reviewers (I.P. and M.C.), with disagreements resolved through discussion and on the basis of the opinion of a third reviewer (F.M.). This review included RCTs published in full or as preprint. Observational non-RCTs were not considered. Studies evaluating both outpatients and inpatients (stratified by the severity of COVID-19) were included.

### 2.3. Types of Intervention

The investigation group included patients with confirmed COVID-19 receiving ivermectin ± standard treatment (ST) as defined in the individual study; the control group included patients receiving ST and/or placebo (Table 1 and Appendix A).

### 2.4. Outcomes

Primary outcomes were overall mortality, progression to severe disease (severe pneumonia, admission to intensive care unit, and/or mechanical ventilation), and serious adverse events. Secondary outcomes included viral clearance and overall occurrence of adverse events. Where available, the outcome measures were reported in different follow-up periods. In studies evaluating prophylaxis, the outcome was efficacy of ivermectin compared to control in preventing COVID-19 among contacts.

### 2.5. Data Collection and Analysis

The following data were extracted by two reviewers (I.P. and M.C.) independently: first author, year of publication, regimens under investigation, outcome measures, and main results. Measures of treatment effect were mean differences (MD) together with 95% confidence intervals (CIs) for continuous outcome measures and risk differences (RD) for binary outcomes. Disagreement was resolved by consensus and by a third reviewer (F.M.), if necessary.

The study weight was calculated using the Mantel–Haenszel method. We assessed statistical heterogeneity using t^2^, Cochran’s Q, and *I^2^* statistics. The *I^2^* statistic describes the percentage of total variation across trials due to heterogeneity rather than sampling error. In the case of no heterogeneity (*I*^2^ = 0), studies were pooled using a fixed-effects model. Where values of *I^2^* were >0, a random-effects analysis was undertaken.

### 2.6. Assessment of Risk of Bias in Included Studies

Two review authors (I.P., M.C.) independently assessed the risk of bias of each study included following the domain-based evaluation described in the Cochrane Handbook for Systematic Reviews of Interventions (Available from www.handbook.cochrane.org, accessed on 18 August 2021) [33,34]. They discussed any discrepancies and achieved consensus on the final assessment. The Cochrane ’risk of bias’ tool addresses six specific domains: sequence generation, allocation concealment, blinding, incomplete data, selective outcome reporting, and other issues relating to bias.

### 2.7. ‘Summary of Findings’ Tables

We used the principles of the GRADE system (Available from www.handbook.cochrane.org, accessed on 18 August 2021) to assess the quality of the body of evidence associated with specific outcomes and constructed ’summary of findings’ tables using REVMAN 5.4 (Review Manager (RevMan) [Computer program]. Version 5.4, The Cochrane Collaboration, 2020) [35]. These tables present key information concerning the certainty of the evidence, the magnitude of the effects of the interventions examined, and the sum of available data for the main outcomes [36]. The GRADE system defines the certainty of a body of evidence as the extent to which one can be confident that an estimate of effect or association is close to the true quantity of specific interest. The certainty of a body of evidence involves consideration of within-trial risk of bias (methodological quality), directness of evidence, heterogeneity, precision of effect estimates, and risk of publication bias.

We present the following outcomes in the ’summary of findings’ table: overall mortality, disease progression, viral clearance, serious adverse events, and rate of infection in prophylaxis studies.

### 2.8. All Calculations Were Made Using REVMAN 5.4

#### Subgroup Analyses

We anticipated heterogeneity in the design and reporting of studies and, to deal with heterogeneity, we planned to carry out subgroup analyses considering specific patients or the characteristics of interventions that may have an effect [37]. Therefore, we considered the following sub-group analyses:-Mortality and disease progression in ivermectin and control group according to baseline clinical conditions (e.g., mild, moderate or severe COVID-19);-Mortality in ivermectin and control group according to the intervention regimen (ivermectin alone or in combination with other active drugs), and according to ivermectin dosage;-Virological data according to the observation period (i.e., 10 and 14 days);-Comparative efficacy of ivermectin in preventing overall COVID-19 and infections stratified by clinical severity.

Once sufficient trials were identified, we planned to carry out a sensitivity analysis comparing the results according to methodological quality (i.e., studies classified as having a ‘low risk of bias’ versus those identified as having a ‘high risk of bias’).

## 3. Results

The search yielded 208 potentially relevant studies (Figure 1, study flow chart).

A total of 180 reports were excluded after preliminary screening; 28 were deemed potentially eligible, and the full-text was assessed. Seventeenstudies were then excluded (eight reviews [2,3,5,6,7,8,15,16], four protocols [37,38,39,40], three non-RCTs [41,42,43], a pharmacokinetic study [44], and a pilot RCT not reporting the predefined outcomes of our review [45]). Hence, 11 studies were available for qualitative synthesis [22,23,24,25,26,27,28,29,30,31,32]. The main features of the studies included are summarized in Table 1 and Appendix A.

Overall, 2436 individuals were enrolled in the 11 RCTs selected for the review: 1295 received ivermectin and 1141 placebo or other treatment. In three studies, ivermectin was given as prophylaxis [24,25,32], and in nine as treatment [22,23,25,26,27,28,29,30,31]. Studies were conducted in Egypt (2); India (1); Argentina (1); Bangladesh (1); Bangladesh and Singapore (1); Spain, Switzerland, and the USA (1); Iraq (1); Iran (1); Colombia (1); and Turkey (1).

### 3.1. Risk of Bias in Included Studies

Four studies (36%) were with high risk of bias for one or more domains, and all the studies were with unclear risk of bias for 1 or more domains (Appendix A).

### 3.2. Allocation

We assessed two studies as being with high risk of selection bias due to the randomization by alternation of the two treatments and because the intervention allocations could have been foreseen in advance [26,30]. The reports of three other studies were unclear for random sequence generation and/or allocation concealment, while five studies (45%) were with low risk of selection biases.

### 3.3. Blinding

*Performance bias*. Four studies (36%) were reported as open label and were graded as high risk of performance bias (blinding of participants and personnel); four studies (36%) were graded as with unclear risk of performance due to the lack of the information required to come to a judgement about ‘high’ or ‘low’ risk of bias related to the blinding of participants and personnel. Three studies were reported as double-blind.

*Detection bias*. No study provided the information required to come to a judgement about ‘high’ or ‘low’ risk of bias related to the blinding of outcome assessors, and were graded as with unclear risk of detection bias.

### 3.4. Incomplete Outcome Data

Two studies were deemed as having unclear risk of attrition bias because of the lack of information on the number of patients that completed the study, or because a high proportion of enrolled patients (10%) did not complete the study. The remaining nine studies (81%) were deemed as having low risk of bias.

### 3.5. Selective Reporting

Selective reporting bias was low in nine studies (81%). One study was deemed to have an unclear risk of bias, and one study with high risk (Appendix A).

### 3.6. Other Potential Sources of Bias

Nine studies were deemed with low risk of other bias, and two studies with unclear risk of other bias.

### 3.7. Effects of Interventions

A summary of the outcomes reported in the included study is provided in Figure 2 (data and analysis). The outcomes most commonly reported were mortality; disease progression; viral clearance; overall side effects; serious side effects; and, for prophylaxis trials, the rate of infection among contacts.

### 3.8. Mortality

Data on mortality were reported in all nine studies evaluating ivermectin treatment. There were 10 deaths out of 796 patients in the ivermectin group compared to 39/791 in the control group (RD −0.02; 95% CIs, −0.05/0.01; *p* = 0.17) (Figure 2).

In eight studies, mortality was reported according to the severity of COVID-19. When the analysis was limited to studies or a subset of patients with baseline mild or moderate disease (8 reports, 1283 patients), there were no differences in mortality between ivermectin and control groups (RD, −0.01; 95% CIs, −0.01/0.00; *p* = 0.26) (Figure 2). The quality of the evidence was deemed low (downgraded for risk of bias and imprecision) (Table 2, summary of findings table).

When the analysis was restricted to studies or subsets of patients with baseline severe diseases (3 reports, 304 patients) [25,26,30], the use of ivermectin decreased mortality compared to controls (RD −0.17; 95% CIs, −0.24/−0.10; *p* = 0.00001). The quality of the evidence was deemed low (downgraded twice for severe risk of selection biases).

In sensitivity analysis, after the exclusion of the study by Elgazzar et al. [35] in the subset of studies with baseline severe conditions the difference in the occurrence of mortality is not longer favouring ivermectin compared to controls (MD, −0.14 (95% CIs, −0.30/0.02; *p* = 0.08). As in the previous analyses, the certainty of the available evidence remains low.

For the mortality outcome, we carried out subgroup analyses considering the drugs included in the ivermectin regimens and the dosage of ivermectin. In three studies (or subset of patients) [23,27,43], ivermectin was given alone and compared to placebo. In seven studies or subsets of patients [25,26,28,29,30,31], ivermectin was given in combination with standard treatment including other active drugs (e.g., doxycycline, azithromycin, remdesivir, steroids, anticoagulants, and others) and compared to controls receiving the same standard treatment. In studies comparing ivermectin to placebo, there was one death (in the control group) out of 467 patients enrolled (RD, −0.00; 95% CIs, −0.02/0.01; *p* = 0.48; moderate quality of evidence; downgraded once for imprecision) (Figure 3).

In studies comparing ivermectin plus standard treatment vs. standard treatment, there were 14 deaths among 681 patients receiving ivermectin, and 53 deaths among 619 patients receiving standard treatment (RD, −0.06; 95% CIs, −0.11/−0.00; *p* = 0.04; low quality of evidence due to risk of bias and inconsistency). Ivermectin, in single dose or 2–5 daily doses, was used as 400 µg/kg dose in four trials [23,25,27,29], and as 200 µg/kg dose in six trials [23,25,29,31,44,45]. In both cases, there was no difference in the mortality rate between ivermectin recipients and control (Appendix A).

### 3.9. Virological Outcomes

The rate of patients with negative RT-PCR test was evaluated at 6–10 days and at 14 days (Figure 4, Table 2).

At day 10, ivermectin did not significantly increase the proportion of RT-PCR-negative patients (RD, 0.10; 95% CIs, −0.12/0.31; *p* = 0.38). The GRADE assessment showed very low quality of evidence due to risk of bias, inconsistency, and imprecision. At day 14, the rate of RT-PCR-negative patients was significantly higher among ivermectin recipients compared to controls (RD 0.21; 95% CIs, 0.05/0.36; *p* = 0.01; low quality of evidence due to risk of bias and inconsistency).

### 3.10. Disease Progression

Disease progression (to severe pneumonia, admission to ICU, and/or mechanical ventilation) was observed less frequently among ivermectin recipients compared to controls (RD, −0.09; 95% CIs, −0.16/−0.02; *p =* 0.01; very low quality evidence; downgraded twice for serious risk of selection biases, and once for inconsistency). The size of the effect was more evident in the high-risk population (severely ill at baseline) (RD, −0.26; 95% CIs, −0.34/−0.17; *p* 0.00001; very low quality of evidence; downgraded twice for serious risk of selection biases, and once for inconsistency) compared to the low-risk population (mild/moderate diseases) (RD, −0.05; 95% CIs, −0.11/0.00; *p* = 0.07; low quality evidence due to risk of bias and imprecision) (Figure 5, Table 2).

### 3.11. Adverse Events

Few participants were reported undergoing a serious event in either Ivermectin or control groups (Appendix A). This comparison was graded as low-quality evidence and downgraded once due to risk of bias and once for imprecision (95% CIs include the line of no effect).

Overall side effects were reported in 253 of the 913 ivermectin recipients and 274 of the 811 controls (RD, −0.01/95% CIs, −0.11/0.08; *p* = 0.78) (Appendix A).

### 3.12. Ivermectin for Prophylaxis among Healthcare and/or Household Contacts of COVID-19

Three studies [24,25,32] evaluated the effectiveness of ivermectin prophylaxis compared to no treatment or to topical treatment with carrageenan (an antiviral compound applied locally in the nasal and oral cavity). In one study [32], clinical evaluation was carried out in all subjects, but RT-PCR only in a minority of patients; hence, asymptomatic infections among contacts may have been missed in both groups, rating down the quality of the evidence for indirectness. The incidence of COVID-19 was significantly lower among ivermectin recipients than controls (RD, −0.28; 95% CIs, −0.33/−0.23; *p* < 0.00001; very low quality of evidence; downgraded for risk of bias, indirectness, and inconsistency (Appendix A)). Ivermectin was significantly more effective than control in preventing mild, moderate, and severe infections (Appendix A).

## 4. Discussion

Other meta-analyses have specifically examined the role of ivermectin for COVID-19 [6,17,18]. One, based on 4 trials and 397 participants, concluded that ivermectin reduced overall mortality and led to a significant clinical improvement compared to the usual therapy, but also highlighted the limitations of these conclusions due to the very low quality of the available evidence [7]. In the other two reviews, overall mortality was the only outcome considered; both studies revealed a significant reduction of the odds of mortality in ivermectin recipients compared to the controls [17,18]. However, there were other methodologic limitations in these two systematic reviews, since no subgroup analysis was carried out to address the heterogeneity observed, and in one review [17], there was no assessment of the methodological quality of included studies. Moreover, the number of studies and participants included in all these reviews was far lower than in our systematic review.

The current review includes 11 RCTs and 2436 participants. Studies were grouped into two main comparisons: (i) ivermectin as treatment of COVID-19; (ii) ivermectin as prophylaxis of COVID-19 in household and/or healthcare personnel contacts with COVID-19 cases. The certainty of the available evidence was quite low or very low, and at best moderate.

Pooled data for ivermectin compared with controls suggest that once the analysis was limited to studies or a subset of patients with baseline mild or moderate disease (8 reports, 1283 patients), there were no differences in mortality between ivermectin and control groups (low level of certainty). When the analysis was restricted to studies or a subset of patients with baseline severe diseases (3 reports, 304 patients), the use of ivermectin significantly decreased mortality compared to controls (low level of certainty). An assessment of low-certainty evidence means that our confidence in the effect estimate is low, and the true effect may be substantially different from the estimate. In subgroup analyses, we found that in studies comparing ivermectin alone to placebo, mortality was comparable: 1 death (in the control group) out of 467 patients enrolled (moderate quality of evidence; downgraded once for imprecision). In studies comparing ivermectin plus standard treatment vs. standard treatment, there were 14 deaths among 681 patients receiving ivermectin, and 53 deaths among 619 patients receiving standard treatment (RD, −0.06; 95% CIs, −0.11/−0.00; *p* = 0.04; low level of certainty due to risk of bias and inconsistency).

The results in terms of disease progression (to severe pneumonia, admission to intensive care unit, and/or mechanical ventilation) were much the same. On average, it is unclear whether or not the use of ivermectin compared to controls decreases disease progression in the low-risk population (very low quality of evidence). These results do not provide a reliable indication of the likely effect, and the possibility that the actual effect will be substantially different is very high. On the other hand, the rate of disease progression was significantly lower in the ivermectin group compared to control, but the quality of the evidence is once again low.

Notably, our data indicate that ivermectin is more active in reducing mortality and clinical progression among severely ill patients, suggesting that the clinical utility of ivermectin may reflect an anti-inflammatory activity of the drug in the late stage rather than an antiviral activity in the early stage of COVID-19. This anti-inflammatory activity has already been demonstrated in animal models of infection and seems to be related to the inhibition of inflammatory cytokines [15,16]. However, our findings should be interpreted with caution due to the low quality of the available evidence.

At day 6–10, it was unclear whether or not the use of ivermectin compared to controls decreased the rate of patients with RT-PCT-negative test (very low quality of evidence). By contrast, at day 14, the rate of patients with negative RT-PCR test was 21% higher (from 5 to 36% higher) among ivermectin recipients, but the quality of evidence was low. An early decrease of viral load in ivermectin group compared to controls was demonstrated in the study by Samaha et al., but the authors reported the cycle threshold values of RT-PCR test and not the rate of patients with a RT-PCR-negative test; therefore, we could not pool these data with those of the studies included in the review

Results from three studies (736 subjects) showed that prophylaxis with ivermectin increased the likelihood of preventing COVID-19 compared to controls (low quality of evidence). Serious adverse events were rarely reported both in ivermectin and controls.

The quantitative analysis conducted in this systematic review has, however, several limitations that do not allow us to draw definite conclusions about the efficacy of ivermectin in this setting. An important limitation is certainly related to the heterogeneity of the studies evaluated, which encompasses both clinical and methodological heterogeneity. Differences between studies in terms of clinical factors, such as setting (i.e., outpatients and hospitalized patients), severity of COVID-19, doses and administration schedule of ivermectin and comparators, and methodological factors such as bias in the selection of patients and in the assessment and reporting of outcomes, were common. Even so, in this systematic review, we have highlighted these differences and addressed heterogeneity using a random effect model and performing subgroup analyses. Another important limitation of this review is that the assessments of ivermectin for COVID-19 continue to be published in preprints and protocol repositories, which do not follow the recommended processes to ensure high quality standards for publications [19]. As much as possible, we have minimized potential biases in the review process. We followed the methods set out in our published PROSPERO protocol, a key feature for providing transparency in the review process and to ensure protection against reporting biases, as well as structuring a summary of findings table and GRADE assessment as required by the new Cochrane standards.

Another important issue to be considered in trials with ivermectin is the drug dosage. Well-controlled dose–response studies need to be considered to carry out a clinical trial of ivermectin. Schmith et al. [45] carried out simulations with the help of a population pharmacokinetic model for predicting total and unbound plasma concentration–time profiles of ivermectin after administration of the approved dose (200 µg/kg,) or much higher doses (60 mg, and 120 mg), in single and repeated doses. According to these results, the IC_50_ value of ivermectin is much higher than the maximum plasma concentration achieved after administration of the above-mentioned three doses of ivermectin [46], and, as a consequence, the chances of success of a trial using the approved ivermectin dose (200 µg/kg) is low. Indeed, after daily dosing of ivermectin 200 μg/kg, lung concentrations are predicted to be around 1/4th of the IC_50__._ On the other hand, it is also conceivable that the in vitro findings do not correlate with in vivo findings, and that concentrations of the drug in lung tissue do not need to reach the IC_50_ for clinical benefit [13,44].

In this review, we carried out a subgroup analysis, according to the dose of ivermectin administered, the approved 200 µg/kg dose, or a 400 µg/kg dose: in neither case were there differences in the mortality rate between ivermectin recipients and controls. There is some evidence that ivermectin is safe, even at higher doses and frequency regimens [46]. Doses of ivermectin such as 120 mg (up to 2000 µg/kg) taken once or at 180 mg (up to 3000 µg/kg) in split doses over 1 week are well-tolerated and safe [47]. It would be important to conduct a well-controlled clinical dose–response study with ivermectin at the approved dose and at a higher dose in relation to placebo in patients with COVID-19. Moreover, future studies might consider new formulations, such as aerosolized and parenteral ivermectin.

The latest version of the WHO living guidance and IDSA guidelines recommends against ivermectin in patients with COVID-19 regardless of disease severity, except in the context of a clinical trial [48,49]. More studies are underway, and it would be premature to conclude that ivermectin has no place in COVID-19 treatment. However, as our systematic review confirms, further evidence is needed to better define potential indications and optimal treatment protocols for ivermectin as a treatment of COVID-19.

## Figures and Tables

**Figure 1 diagnostics-11-01645-f001:**
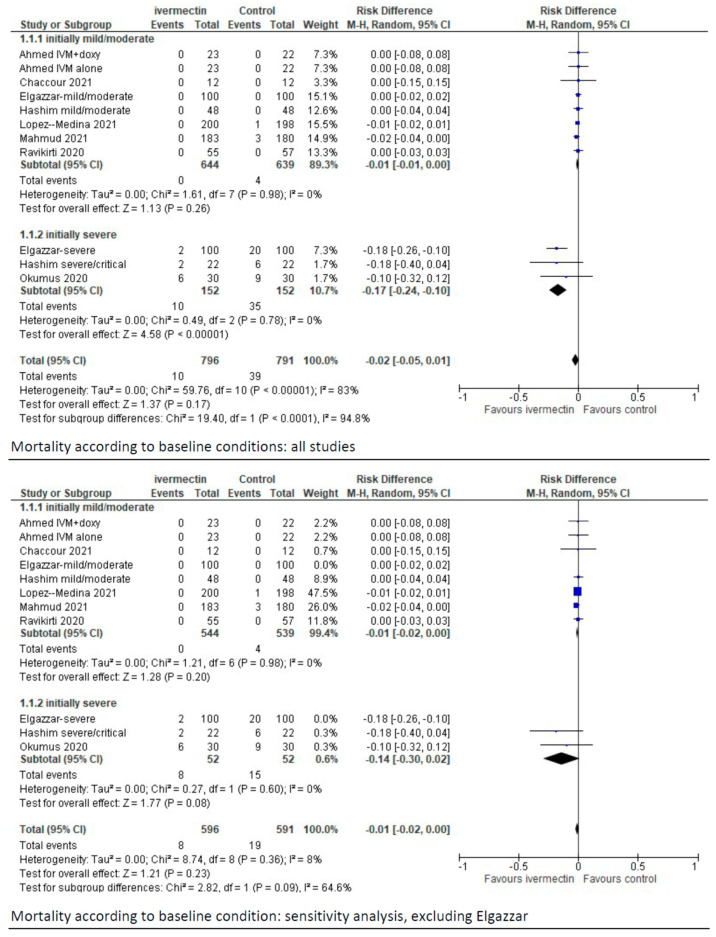
Forest plot of the comparison. Outcome: mortality according to baseline conditions. Top: all studies; bottom: sensitivity analysis excluding Elgazzar et al. [25].

**Figure 2 diagnostics-11-01645-f002:**
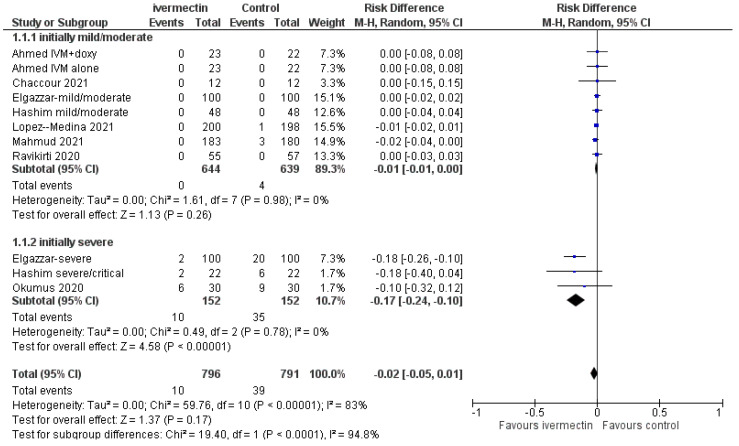
Forest plot of comparison. Outcome: overall mortality according to initial clinical status.

**Figure 3 diagnostics-11-01645-f003:**
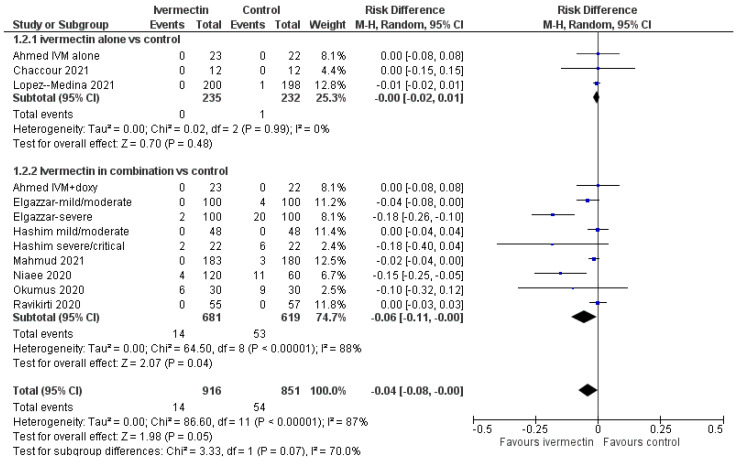
Forest plot of comparison. Outcome: mortality according to ivermectin regimens (ivermectin alone or in combination with ‘standard treatment’).

**Figure 4 diagnostics-11-01645-f004:**
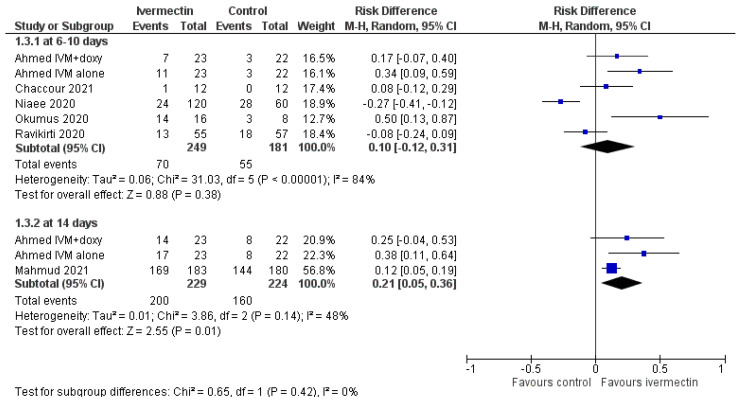
Forest plot of comparison. Outcome: rate of patients with RT-PCR test for COVID-19 negative at days 6–10 and at day 14.

**Figure 5 diagnostics-11-01645-f005:**
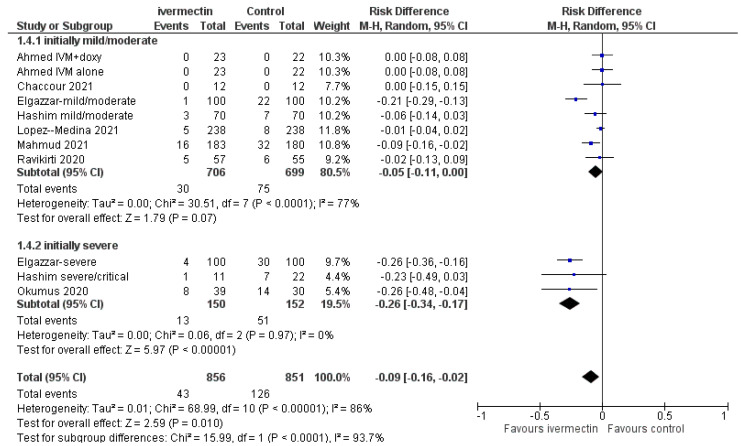
Forest plot of comparison. Outcome: clinical progression according to initial clinical status.

**Table 1 diagnostics-11-01645-t001:**
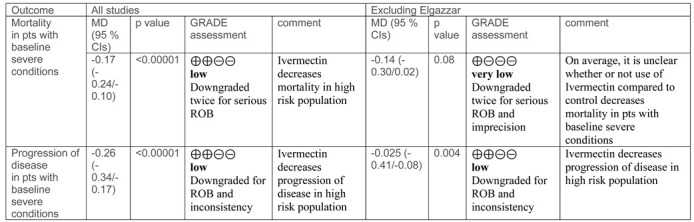
Sensitivity analysis of some outcomes of the meta-analysis.

MD, mean difference. CIs. Confidence intervals. ROB, Risk of bias.

**Table 2 diagnostics-11-01645-t002:** Ivermectin compared with control intervention for COVID-19 treatment or prevention. Patient or population: Patients with COVID-19 for treatment studies; healthcare personnel and household contacts for prevention studies. Settings: outpatients and hospitalized patients. Intervention: ivermectin ± standard treatment. Comparison: standard treatment.

Outcomes	Illustrative Comparative Risks * (95% CI)		Relative Effect (95% CI)	No of Participants (Studies)	Quality of the Evidence (GRADE)	Comments
	Assumed Risk	Corresponding Risk				
	Control	Ivermectin				
Mortality according to baseline conditions	**Low-risk population (mild/moderate disease)**		**RD −0.01 (−0.01/0.00)** **RD −0.17 (−0.24/−0.10)**	**1283 (7)** **304 (3)**	⊕⊕⊝⊝ **low ^1^** ⊕⊕⊝⊝ **low ^2^**	On average, it is unclear whether or not use of ivermectin compared to control decreases mortality in low-risk population. The average benefit is higher in the high-risk population.
	mortality ranged from 0 to 1.6%	mortality was 1% lower (from 0 to 1% lower)				
	**High risk population (severe disease)**					
	mortality ranged from 20% to 30%	mortality was 17% lower (from 10% to 24% lower)				
Viral clearance (% patients)	**At 6–10 days**		**RD 0.10 (−0.12/0.31)** **RD 0.21 (0.05/0.36)**	**430 (5)** **360 (2)**	⊕⊝⊝⊝ **very low ^3^** ⊕⊕⊝⊝ **low ^4^**	On average, it is unclear whether or not use of ivermectin compared to control decreases rate of patients with RT-PCR negative test after 6–10 days. After 14 days, ivermectin increases rates of pts with negative RT-PCR test compared to control.
	rate of patients with negative RT-PCR ranged from 0 to 46.6 per 100	rate of patients with negative RT-PCR was 10% higher (from 31% higher to 12% lower)				
	**At 14 days**					
	rate of patients with negative RT-PCR ranged from 36 to 80 per 100	rate of patients with negative RT-PCR was 21% higher (from 5% to 36% higher)				
Disease progression (severe pneumonia, admission to intensive care unit, and/or mechanical ventilation) according to baseline conditions	**Low-risk population (mild/moderate disease)**		**RD −0.05 (−0.11 to 0.00)** **RD −0.09 (−0.16/−0.02)**	**1405 (7)** **302 (3)**	⊕⊝⊝⊝ **very low ^3^** ⊕⊕⊝⊝ **low ^4^**	On average, it is unclear as to whether or not use of ivermectin compared to control decreases disease progression in the low-risk population. The average benefit is higher in the high-risk population.
	disease progression ranged from 0 to 22 per 100	rate of patients with disease progression was 5% lower (from 0 to 11% lower)				
	**High-risk population (severe disease)**					
	disease progression ranged from 30 to 46 per 100	rate of patients with disease progression was 9% lower (from 16 to 2% lower)				
Serious adverse events	serious adverse events ranged from 0 to 2.5 per 100	serious adverse events were 1% higher (from 1% lower to 2% higher)	**RD 0.01 (−0.01/0.02)**	**1428 (6)**	⊕⊕⊝⊝ **low ^1^**	Serious adverse events were rarely reported in both ivermectin and control groups.
Prevention of infection in healthcare and household contacts of COVID-19 pts	rate of infection ranged from 10 to 58 per cent	rate of infection was 28% lower (from 61 to 41% lower)	**RD −0.28 (−0.33/−0.23)**	**736 (3)**	⊕⊝⊝⊝ **very low ^5^**	Prophylaxis with ivermectin increased the likelihood of preventing COVID-19 compared to controls.

* The assumed risk is the mean control group risk across studies. The corresponding risk (and its 95% confidence interval) is based on the assumed risk in the comparison group and the relative effect of the intervention (and its 95% CI). CI: confidence interval; RD: risk difference. GRADE: working group grades of evidence. High quality: further research is very unlikely to change our confidence in the estimate of effect. Moderate quality: further research is likely to have an important impact on our confidence in the estimate of effect and may change the estimate. Low quality: further research is very likely to have an important impact on our confidence in the estimate of effect and is likely to change the estimate. Very low quality: we are very uncertain about the estimate. ^1^ Downgraded for risk of bias and imprecision (95% CI includes line of no effect); ^2^ downgraded twice for risk of bias; ^3^ downgraded for risk of bias, imprecision, and inconsistency (due to heterogeneity); ^4^ downgraded for risk of bias and inconsistency; ^5^ downgraded for risk of bias, inconsistency, and indirectness.

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
