# Peer review of "Ivermectin for Prophylaxis and Treatment of COVID-19: A Systematic Review and Meta-Analysis"

_diagnostics, 2021, doi:10.3390/diagnostics11091645_

Round 1

Reviewer 1 Report

This is a timely systematic review (SR), and is much needed. As we are all aware, there are many SRs about the same topic, some of which have been recently published and some are ongoing. Expectedly, more will come in the near future as well.

The authors did a good job at searching the available ones. However, they missed some studies. Therefore, if the other studies that appeared during the past 3 months.

Some of these include:

J Pharm Pharm Sci. 2020;23:462-469. doi: 10.18433/jpps31457. Pharmacol Rep. 2021 Jun;73(3):736-749. doi: 10.1007/s43440-020-00195-y Trials. 2020 Jun 8;21(1):498. doi: 10.1186/s13063-020-04421-z.   Also, I am surprised the following study was not used although it was published during the time used bu authors for inclusing clinical trials. So, this ought to be included as well, despite the fact that it is a pilot study. Viruses. 2021 May 26;13(6):989. doi: 10.3390/v13060989.   Moreover, it would be nice to highlight the earlier work that elaborated on the repurposing of ivermectin (see  Front Immunol. 2021 Mar 30;12:663586. doi: 10.3389/fimmu.2021.663586. eCollection 2021.)   In addition, a year ago, Kaddoura et al published a nice piece that highlighted various drugs unders consideration for COVID. There, ivermectin received important attention  (Front Pharmacol. 2020 Aug 6;11:1196. doi: 10.3389/fphar.2020.01196. eCollection 2020.)   Was there any gender differences i the respnsiveness to ivermectin among COVID patients?    

Author Response

We thanks the reviewers for the usefull suggestions. The reviewer suggests to add 5 studies to the manuscript. 2 of these has been already considered and cited (Padhy, reference no.6, and Chaccour, ref. 26). As suggested, we add 2 new references (Khaddoura and Wehbe, references 4 and 7) and the RCT by  Samaha. Unfortunately the study by Samaha et al did not report the outcomes we have prespecified in our protocol, se we were not able to include this study in the quantitative abalysis. However, we have specified in the discussion (see lines 310-322) the virologic reponse in ivermectin group, reported as Ct values of the PCR test (and not as rate of patients with negative PCR test, which is the endpoint we have extracted from the other RCTs included in the current analysis).

Finally, the reviewer ask if  there was any gender differences in the responsiveness to ivermectin among COVID patients. Unfortunately we were not able to extract this information from the available studies

Reviewer 2 Report

The paper focused on the topic of ivermectin for treatment and prevention of COVID-19 which is very important to those country which could not afford expensive medication such as remdesiver or tocizilumab etc. Unfortunately the clinical trials collected were too heterogenous and could not support the use of ivermectin for treatment and prevention of COVID-19.

Too many duplication of results in the beginning of discussion.  The discussion should focus on the comparison between different reviews of ivermectin. Yet the review paper is quite well design and prepared for publication.

Author Response

Following the reviewer suggestion, we have modified the discussion starting with the results of the available systematic reviews, and then showing strenghts and limitations of our review

Reviewer 3 Report

The well-written, thorough paper research and analyses manuscript conclude against ivermectin in patients with COVID-19 no matter the disease severity is.No changes are recommended

Author Response

thank you